# Effects of chemotherapy on skeletal muscle mitochondrial oxidative capacity using near-infrared spectroscopy (NIRS): Protocol paper for an observational mixed model repeated measures design in patients with breast and gynecological cancer

Randolph Edward Hutchison[1]◉, Shannon Smith[2]‡, Chloe Caudell[2]‡, Sara Biddle[3]◉, William Larry Gluck[3]◉, Jennifer L. Trilk[2]◉*

**1** Department of Health Sciences, Furman University, Greenville, South Carolina, United States of America, **2** Department of Biomedical Sciences, University of South Carolina School of Medicine Greenville, Greenville, South Carolina, United States of America, **3** Cancer Institute, Prisma Health, Greenville, South Carolina, United States of America

◉ These authors contributed equally to this work.
‡ SS and CC also contributed equally to this work.
* trilk@greenvillemed.sc.edu

## Abstract

Mitochondrial dysfunction, a hallmark of metabolic disturbances in the skeletal muscle, has previously been studied in health participants using invasive muscle biopsy and/or time consuming, high-cost magnetic resonance spectroscopy. However, less is understood regarding mitochondrial dysfunction in patients with cancer. Near infrared spectroscopy (NIRS) is a non-invasive and cost-effective approach to assessing mitochondrial function of skeletal muscle by measuring oxygenated and deoxygenated hemoglobin and calculating the resulting tissue saturation index. NIRS has not yet been utilized to evaluate skeletal muscle change in cancer patients throughout chemotherapy. Therefore, we plan to conduct a single center clinical trial, using an observational mixed model repeated measures design to evaluate the change in mitochondrial oxidative capacity from baseline and throughout progression of an oncologist-prescribed chemotherapy regimen. Evaluation of mitochondrial function will be performed by taking real-time, NIRS in-situ measurements within the working muscle during stationary cycling exercise and subsequent recovery periods (e.g., "on" kinetics and "off" kinetics). We also plan to observe if there is a difference in mitochondrial oxidative capacity between different chemotherapy regimens across different patients. This pilot study will provide information on the feasibility of capturing on/off kinetics mitochondrial function data longitudinally using NIRS in patients newly diagnosed with breast or gynecological cancer, provide preliminary data

**Data availability statement:** No datasets were generated or analysed during the current study. All relevant data from this study will be made available upon study completion.

**Funding:** This study was partially funded through the Prisma Health Cancer Institute Office of Philanthropy and Partnership (https://academics.prismahealth.org/research-and-innovation/division-of-research) and the University of South Carolina School of Medicine Greenville Summer Scholars Program (https://sc.edu/study/colleges_schools/medicine_greenville/research/student_research_opportunities/index.php). RH, JT, SS, CC, WLG received these funds. The sponsors and funders played no role in the study design, data collection and analysis, decision to publish, or preparation of the manuscript.

**Competing interests:** The authors have declared that no competing interests exist.

for future extramural funding, as well as inform the scientific community of results through dissemination via conferences and peer-reviewed journal publications.

This clinical trial has been registered with clinicaltrials.gov (Identifier: NCT006672497). Data collection started on July 26, 2021 and is ongoing through March 27, 2026. The authors confirm that all ongoing and related trials for this observation trial (no drug or intervention) are registered.

## Introduction

In the United States, more than 3.8 million women are currently receiving chemotherapy treatment, or have completed treatment, for breast or gynecological cancer [1]. Fortunately, over the past 25 years, the ability to detect and treat cancer has allowed breast cancer 5-year survivor rates to increase to 85 [1], ovarian cancer 5-year survivor rates to increase to 49% [2], and uterine cancer 5-year survivor rates to increase to 81% [3]. Unfortunately, while chemotherapy has improved survival rates amongst patients with breast and gynecological cancer, side effects include unfavorable body composition changes [4], increases in fatigue [5], reductions in physical functioning [6], and decreased quality of life (QOL) [7]. Many of these symptoms may be due to post-chemotherapy cachexia [8–10]. According to the National Cancer Institute, cachexia is estimated to occur in up to 80% of people with advanced cancer [8,11] and involves reduced size and content of mitochondria [12,13] and subsequent deterioration of mitochondrial function and its oxidative capacity associated with reduced energy production for physical functioning [9,10,14]. Cachexia has negative implications for patients with breast and gynecologic cancer; therefore, the increase in survivorship has been an increase in focus on understanding mitochondrial dysfunction, with the aim to understand how to improve quality of life for [10].

Previous studies have relied on more invasive tools to assess pathological progression of mitochondrial [15,16]. Muscle biopsies have been the gold standard assessment but can cause significant discomfort to the patient [17]. Additionally, muscle biopsies may not represent *in vivo* activation of the skeletal working muscle tissue associated with [17]. Magnetic Resonance Spectroscopy (MRS) provides *in vivo* measurements but is cost prohibitive, time intensive, requires technical expertise, and competes with other clinical needs within the [17]. Metabolic ventilatory measures have been established as a gold standard in evaluation of pathophysiology with many chronic diseases except for large scale studies in cancer chemotherapy treatment [18–20]. These measurements could provide critical mechanistic understanding of oxygen delivery and utilization from a systematic whole-body approach [21]. To understand local skeletal muscle oxygen uptake kinetics, Near Infrared Spectroscopy (NIRS) has been validated with muscle biopsies and [22,23], as a low cost, non-invasive alternative assessing mitochondrial capacity in clinical [17]. NIRS measurement of tissue oxygenation has been well studied and a thorough background on NIRS can be found elsewhere [17,22,24,25]. Briefly, biological tissue has a relatively

good transparency for light in the near infrared region (700–1300 nm) and transmits photons through organs for *in situ* monitoring. The refracted light is converted into grams of oxyhemoglobin ($O_2Hb$) and deoxyhemoglobin (HHb) based on the modified Beer-Lambert law [26]. NIRS devices with multiple transmitters at different distances from the receiver can calculate the Tissue Saturation Index (TSI)=$O_2Hb/(O_2Hb + HHb)$ and reflects the percentage of measured blood that is oxygenated.

Assessment of mitochondrial capacity with NIRS includes studies of clinical populations with peripheral artery [27], chronic kidney [24], and neurologic diseases such as traumatic spinal cord injury and multiple [22], as well as comparisons with healthy [28] and age-related differences in skeletal muscle oxidative [17]. However, there is a dearth of NIRS studies examining the effects of chemotherapy on skeletal muscle mitochondrial function in patients with cancer. Only two studies to our knowledge have utilized NIRS to measure muscle oxygenation variables in cancer survivors. Ederer, et al. [29] determined that [HHb] and [Hb]total were significantly lower in eight cancer survivors (all eight received chemotherapy and five received radiation) compared to healthy controls during exercise at ventilatory threshold. Grigoriadis et al. [30], did not find a significant difference in muscle oxygenation variables between 23 patients with breast cancer (16 received chemotherapy, nineteen received radiation therapy, and 21 received hormonal therapy) and 23 participants who served as age and BMI matched controls. Recent literature of near infrared spectroscopy related to cancer primarily involves detection and identification of cancerous [31] or the use of near infrared light as a photoimmunotherapy [32]. Therefore, the hypothesized loss of mitochondrial capacity throughout chemotherapy treatment has not been studied longitudinally starting at baseline and tracking through the chemotherapy regimen in breast and gynecologic cancer patients with NIRS to date.

The aims of our study are to develop a protocol that 1) assesses mitochondrial dysfunction in cancer patients using exercise that is tolerable for the patients throughout the entire chemotherapy regimen and 2) observe the changes in mitochondrial functional capacity between different chemotherapy regimens for each chemotherapy cycle. The protocol also will help us evaluate a potential correlation between subjective self-reported fatigue and perceived effort with corresponding objective whole-body metabolic measurements and NIRS measurements. The proposed study will help inform clinical decision makers in the oncology community regarding current chemotherapy regimens (based upon the National Comprehensive Cancer Network, NCCN) that patients may best tolerate for cachexia and fatigue.

## Materials and methods

### Organization and conduct

The study protocol (Fig 1) was approved in July of 2021 by the Prisma Health Office of Human Subjects Protection Institutional Review Board Committee C (Protocol # Pro00110826). This study is sponsored through the Prisma Health Cancer Institute Office of Philanthropy and Partnership, the University of South Carolina School of Medicine Greenville Summer Scholars Program, and Furman University. The study sponsors and funders, played no role in study design; collection, management, analysis, and interpretation of data; writing of the report; and the decision to submit the report for publication, including whether they will have ultimate authority over any of these activities. This clinical trial has been registered with clinicaltrials.gov (Identifier: NCT006672497). Data collection started on July 26, 2021 and is ongoing through March 27, 2026. The authors confirm that all ongoing and related trials for this observation trial (no drug or intervention) are registered.

### Experimental design

We will use an observational mixed model repeated measures design to longitudinally compare (pre-chemotherapy and throughout treatment) patients' skeletal muscle mitochondrial capacity of the vastus lateralis during stationary cycling. ANOVAs will be used to assess this same mitochondrial capacity between chemotherapy regimens at each chemotherapy cycle. Pearson or spearman correlations will be used to assess relationships between self-reported fatigue and effort

| TIMEPOINT** | Enrollment | Post Allocation | | | | | Close-out |
|---|---|---|---|---|---|---|---|
| | -t1 (with Oncology Research Nurse) | t1 (Lactate Threshold Baseline Testing in Human Performance Lab) | t2 (baseline on-off kinetics test, pre-chemotherapy cycle 1) | t3 (pre-chemotherapy cycle 2) | t4 pre-chemotherapy cycle 3 | Etc. until chemotherapy cycles complete | Post Test |
| **ENROLLMENT:** | | | | | | | |
| Eligibility screen | x | | | | | | |
| Informed Consent | x | | | | | | |
| -see inclusion/exclusion criteria (Table 1) | x | | | | | | |
| -feasibility: capable, willing to participate | x | | | | | | |
| -Godin Leisure Survey | | x | | | | | |
| -Ultrasound for adipose tissue thickness | | x | | | | | |
| -Mini Mental State Exam | | x | | | | | |
| Allocation (N/A, no intervention, observational trial only) | | | | | | | |
| INTERVENTIONS: Observational study, no interventions, gathering assessments throughout chemotherapy based on oncology prescribed, chemotherapy standard of practice | | | | | | | |
| **ASSESSMENTS:** | | | | | | | |
| **[Baseline variables]** | | | | | | | |
| Global Heath Promis Survey | | x | x | x | x | x | x |
| Brief Fatigue Inventory | | x | x | x | x | x | x |
| Physical Activity | | x | x | x | x | x | x |
| **[Outcome variables]** | | | | | | | |
| NIRS Data (O2Hb, HHb, TSI, $\tau$, etc.) | | x | x | x | x | x | x |
| Ventilatory Metabolic Measurements (VO2, VCO2, etc.) | | x | x | x | x | x | x |
| Heart Rate | | x | x | x | x | x | x |
| Rate of Perceived Exertion | | x | x | x | x | x | x |
| **[Other data variables]** | | | | | | | |
| Height | | x | x | x | x | x | x |
| Weight | | x | x | x | x | x | x |
| Blood Pressure | | x | x | x | x | x | x |
| Temperature | | x | x | x | x | x | x |
| Ultrasound for adipose tissue thickness | | x | x | x | x | x | x |

**Fig 1. Schedule of enrollments, interventions, and assessments.**

values and metabolic measurements (whole-body as well as NIRS). The study population will include female patients with breast or gynecological cancer. All testing will take place in the University of South Carolina School of Medicine Greenville's Human Performance Laboratory in the Prisma Health Cancer Institute.

## Sample size

We conducted a power analysis using Monte Carlo data simulation. We conducted the analysis using a mixed effects model with a fixed effect of chemotherapy cycle (time) and random effects for slope and intercept by participants with unstructured correlation [33–36]. Cheng [35] et al. suggest the use of unstructured correlation when time is used as a fixed effect (chemotherapy cycle) and we are not making specific assumptions about the shape of the curve. Due to limited studies of NIRS measurements of patients with cancer, we used the conservative estimate of average recovery kinetics time constant (τ) of patients with chronic heart failure (25.35 seconds) and extrapolate average change in τ throughout chemotherapy from Foulkes using results from Beever [37,38]. We assume that the random intercept will follow a gamma distribution with shape 1.57, a scale of 12.5, and a right shift of 13.25. Additional assumptions from Foulkes [38] and Beever [37] include the random slope is assumed to follow a normal distribution with an extrapolated standard deviation, as well as a normal distribution of the error with standard deviation of 5.54 [39,40]. Analysis indicates that using a linear mixed effects model with sample size of at least n = 30 will be able to detect an effect of chemotherapy on τ with a power of 0.8. Due to potential dropout or missing data, data collection will continue until data is collected on 30 subjects who have completed the study. As the effect of chemotherapy type on τ is unknown and unable to be approximated with data simulation, we are unable to conduct a full power analysis with the effect of chemotherapy type on τ [29,41,42]. Based on treatment regimens of pilot data, chemotherapy type can be divided into cytotoxic (platinum-based, anthracyclines, etc.) for 3–4 cycles and non-cytotoxic such as immunotherapy for an additional 3–4 cycles. In some cases, treatment would be given concurrently and could be treated as a separate group of cytotoxic + non-cytotoxic. We assume that at least n = 8 of each chemotherapy type will be necessary to detect any differences in trajectory of τ over time consisting of 3 or more cycles of chemotherapy Comparisons between drug regimens could be adjusted based on updates to the NCCN guidelines.

## Participants and eligibility requirements

Participants recruited for this study will be female patients with breast cancer or gynecological cancer who will undergo a standard of care chemotherapy treatment as directed by their oncologist. Participants are all women who are undergoing adjuvant therapy, meaning that gynecological and breast cancer patients are not presenting clinical evidence of disease (non-metastatic) therefore can be considered from the same population for modeling purposes. Additionally, treatment regimens for these patient populations are similar based on NCCN guidelines which warrants modeling them together. Participants who are cleared by their oncologist to participate will be recruited through informative pamphlets and by oncology nurses trained in research and human subjects protection in Prisma Health System's Cancer Institute at Greenville Memorial Hospital, as well as Prisma Health affiliated oncologists' offices. Participants will be informed of the study's purpose, availability, timeline, and potentially associated minimal risks. Study staff will then obtain written informed consent from participants who agree to participate and before any data collection. Patient inclusion and exclusion criteria are listed in Table 1.

## Observational protocol for first and subsequent visits

Before starting chemotherapy, patients will undergo baseline testing that includes a lactate threshold test and an ON-OFF kinetics exercise test. The lactate threshold test will determine the power output (110% power at lactate threshold, $P_{LT+}$) to be used for the follow-up baseline on-off kinetics test (ON-OFF). The ON-OFF kinetics test will take place at least 24

**Table 1. Inclusion and exclusion criteria.**

| Inclusion Criteria |
| --- |
| Female patients diagnosed with breast cancer or gynecological cancer without distant metastasis |
| Age > 20 years old |
| Able to perform exercise on a stationary cycle ergometer at moderate intensities for a maximum of 15 minutes |
| Hemoglobin (Hb) values > 10 g/dl at baseline |
| Alanine aminotransferase (ALT) and Aspartate transferase (AST) values less than 2.5X the upper limit of normal by institutional standards |
| Godin-Shephard Leisure Time Physical Activity Questionnaire (GLTEQ) score of 14 or higher |
| **Exclusion Criteria** |
| Metastatic breast or gynecological cancer |
| Clinically advanced cardiovascular disease or pulmonary disease or disease requiring continuous oxygen supplementation |
| Greater than 2 centimeters of subcutaneous adipose tissue on the anterior thigh |
| Inability to walk or stand, movement disorders, spinal cord injuries |
| Autoimmune disorders |
| Pregnant or breastfeeding |
| Mini-Mental State Examination (MMSE) score < 24 |
| History of chemotherapy within 5 years prior to beginning participation |

hours after the lactate threshold test and consist of 3 cycles of 2 minutes at $P_{LT+}$ and 2 minutes of complete rest. Whole body $O_2$ uptake and $CO_2$ production will be captured via a metabolic cart during each test. Testing will take no more than 60 minutes per session, with a maximum study total of 300 minutes (about 5 hours) of participation per patient. Patients will be requested to avoid exercise and use of certain substances (tobacco, alcohol, caffeine, and contraindicated medications) 24 hours before testing. If laboratory baseline evaluation cannot be completed prior to the participant's treatment initiation, treatment will not be delayed, and patients will be waived from the study.

The independent variables in this study are drug regimen and number of chemotherapy treatments. The dependent variables in this study will be the changes in $O_2Hb$, $HHb$, and TSI of the vastus lateralis, indicative of mitochondrial oxidative capacity after a bout of cycling. The specific variables measured will be the average recovery kinetics time constant, $\tau$, and TSI. Indirect calorimetry (O2 uptake, CO2 production) also will be captured at rest and during the ON/OFF kinetics as control variable comparisons for whole-body O2 uptake and utilization.

**Instruments and equipment.** The following devices will be utilized throughout the study: NIRS PortaMon by Artinis (Einsteinweg, Netherlands), Lode Corival (Groningen, Netherlands) stationary bike ergometer, Parvo Medics TrueOne 2400 (Salt Lake City, Utah) metabolic cart, Polar FT7 heart rate (HR) monitor (Kempele, Finland), Lactate Plus analyzer (Edina, Minnesota) and Sonosite Titan Portable Ultrasound System (Bloomfield, Connecticut) for adipose tissue thickness measurement.

**Fatigue and quality of life surveys.** We will administer the Mini-Mental State Exam (MMSE) [43–45], Brief Fatigue Inventory (BFI) [46], Physical Activity Intake [47,48], PROMIS Global Health [49,50], and GLTPAQ surveys [51] prior to participant engagement in cycling exercises at the beginning of the first visit. We will administer BFI, PROMIS Global Health, and Physical Activity follow-up surveys at the beginning of each individual study session to assess any changes in cancer related fatigue associated with chemotherapy treatment that may develop throughout the study period.

**Visit #1: Lactate threshold test to tetermine power output, NIRS orientation.** Participants will be asked to wear comfortable exercise clothes with shorts or equivalent so that the NIRS device can be placed directly on the skin. Participants will enter the performance lab and sit at rest in a chair. The NIRS device will be secured to the participant's leg at a standardized measurement of ⅔ distance along a line from the anterior superior iliac spine to the lateral border

of the patella [52]. This distance will be measured and marked medially and laterally on the patient's leg at a place that will not disrupt signal transmission. At this marked location, an ultrasound probe will be used to measure the subcutaneous adipose tissue thickness (ATT) between the epidermal layer and the muscular layer. If the ATT meets the inclusion/exclusion criteria, the NIRS device will be placed on the vastus lateralis. The ultrasound ATT measurement will be recorded. The NIRS device will be secured to the participant's leg via double-sided non-latex adhesive tape applied between the device and the leg and lightly secured in position to the skin via non-latex tape. Pictures of the anatomical location of NIRS device placement with a reference tape measure will be taken to standardize placement at every visit.

The patient will then complete the following questionnaires: contact information form, GLTPAQ, MMSE, BFI, PROMIS, and Physical Activity Form. After the completion of the questionnaires, study staff will review the GLTPAQ and Physical Activity form. Study staff will determine the initial power wattage for the test based on the answers to these surveys.

Sitting with both feet on the floor and legs uncrossed, manual blood pressure will be obtained two separate times, five minutes apart after 10 minutes of seated rest. Blood pressure measurements with an average of <140/<90 is to be considered safe to continue with testing [20]. After the blood pressure measurement is obtained, the patient will have their temperature measured one time and their height and weight measured two times without shoes. The HR monitor will be fitted on the patient to continuously monitor HR throughout exercise. The rate of perceived exertion (RPE) scale will be explained to the patient. The patient will mount the stationary bike and study staff will help adjust the seat, handlebars, and pedals to the comfort of the patient. Study staff will then place a fitted silicone one-way rebreathing mask on the patient to measure their metabolic data using indirect calorimetry via the metabolic cart that captures expired gasses ($O_2$ and $CO_2$). Capillary blood lactate levels will be obtained by finger stick for the lactate analyzer.

Participants will sit on the stationary bike for two minutes to capture expired gases and blood lactate levels at rest. At the end of rest, the patient will begin pedaling at a self-selected steady cadence between 60 and 100 RPMs at the initial power wattage determined by study staff. The patient will pedal at this cadence for the remainder of the test. Every three minutes, the power output will increase by 15 Watts. With thirty seconds remaining in each stage, blood lactate levels will be measured in duplicate and study staff will prompt the participant to quantify their effort using the RPE scale. When the duplicate blood lactate measurements exceed 4.0 mmol/L, the participant will complete the current stage and end the test.

After completion of the test, study staff will calculate the power output that is equivalent to 1.1x the wattage at which patients' lactate exceeded 4.0 mmol/L ($P_{LT+}$). This power output will be used as the exercise intensity for subsequent on-off kinetics testing.

**Visit #2+: On-off kinetics testing.** Using the reference photo taken at the first visit, mark the NIRS device location and use the ultrasound to measure and record ATT. If the ATT measurement matches the ATT from the first visit, secure the NIRS device with tape as previously described. The participant will fill out the BFI, PROMIS Global Health, and Physical Activity Follow Up forms and update contact information if needed.

The participants will sit on the bike for a two-minute rest. Participants will begin pedaling for a warmup at 0 Watts for 2 minutes at a self-selected steady cadence of 60–100 RPM. After the 2-minute warm up, the power output will increase to $P_{LT+}$. Participants will cycle at the same self-selected cadence for 2 minutes ("on-kinetics''). Thirty seconds before the end of the on-kinetics stage, participants will be asked their RPE. At the end of the 2-minute on-kinetics period, participants will stop pedaling and study staff will remove their non-measured leg from the pedal to rest on a box in a relaxed, extended position to aid in total relaxation and remove any additional movement of the measured leg. During the rest period, participants will remain in recovery sitting on the bike for 2 minutes ("off-kinetics'' stage). Participants subsequently will start pedaling at their self-selected cadence for their next 2 minute "on-kinetics" stage, followed by another 2-min "off-kinetics'' period. This cycle of exercise followed by rest, referred to as on-off kinetics, will repeat for a total maximum of 3 times, or unless the patient requests to stop.

## Outcome measures

**Collection of biometrics at each visit.** Study staff will measure O2Hb, HHb, and TSI, via the NIRS PortaMon device during the on-off kinetics protocol. We also will measure ventilatory whole-body components including $O_2$ and $CO_2$ via the Parvomedics metabolic cart. Participants' perceived exertion to the exercise will be measured via RPE during every exercise stage.

NIRS data will be collected using Artinis Medical System's Oxysoft software at a frequency of 10hz. A differential path-length factor (DPF) value of 4 will be selected, as recommended by Artinis (Einsteinweg, Netherlands). The PortaMon will be placed on the vastus lateralis of participants for a period of at least 10 minutes prior to beginning exercise in accordance with the recommendation of Cortese L, et al. to allow calibration of the PortaMon [53].

Throughout data collection, researchers will closely observe for excess movement from participants that may affect the data collection. These sections will be removed from analysis. After data collection, a moving average with a filter width of 1.5 seconds will be applied to smooth leg movement artifacts that occurred during cycling. Data cleaning and analysis will be conducted using R (R Core Team, 2020] and RStudio (Rstudio Team, 2020].

**Data analysis and statistical analysis.** The following values from the NIRS device will be used to assess both aims: 1) mitochondrial capacity change over time (chemotherapy cycle) and 2) between drug regimens. Previous literature has shown that oxygenation levels at rest after moderate and/or high intensity exercise or after occlusion follows an asymptotic curve of the form [54]:

$$y = A + (B_0 - A)\, e^{-\tau/t} \tag{1}$$

Where $y$ is the oxygenation at time $t$, $A$ is the oxygenation as $t$ approaches infinity, $t$ is time measured in seconds, $B_0$ is the value at $t=0$, and $\tau$ is the time constant. For each rest period, we will run an asymptotic regression for O2Hb, HHb, and TSI by time in seconds to find the coefficient estimate for $\tau$.

Each data set will contain three rest periods; therefore, each data set will have three $\tau$ values for each variable. We will average the $\tau$ values to obtain one average rest $\tau$ for $O_2$Hb, HHb, and TSI per data set. Since continuous wave NIRS devices measure oxy- and deoxyhemoglobin as changes from baseline, raw oxy- and deoxyhemoglobin values will have no meaning and are not comparable except when measured as changes over time or as $\tau$ values.

We will analyze the $\Delta O_2 Hb$, $\Delta HHb$, and $\Delta TSI$, calculated as the change in steady state levels of $O_2$Hb, HHb, and TSI respectively from work to rest, with steady state values calculated as the mean value of the final thirty seconds of the stage. For each data set of one on-off kinetics, there will be three work/rest period pairs therefore we will calculate three delta values and average the three delta values to obtain one delta value for $O_2$Hb, HHb, and TSI per data set. The three work/rest transitions will allow for reliability testing with intraclass correlation coefficients. For participant data to be included in the final analysis, the participant will need to complete at least three on-off kinetic laboratory visits. Each participant will complete an on-off kinetics at least two weeks apart, but not more than one month apart in accordance with their prescribed chemotherapy treatment regimen. Each on-off kinetics will correspond with the first infusion of a chemotherapy cycle. To allow participants a maximum amount of time to recover from infusions before participating in exercise, the corresponding on-off kinetics takes place within 48 hours of the first infusion of the subsequent chemotherapy cycle. Since chemotherapy treatments vary depending on oncologist recommendation, each participant may complete different lengths of treatment, resulting in each participant having different numbers of data points. This will prevent us from using traditional statistical analysis such as repeated measures ANOVA. Since it is not time itself, but rather the compilation of chemotherapy infusions over time that we predict will cause changes in measures of mitochondrial oxidative function, we will treat the number of chemotherapy cycles as an ordinal variable in this analysis. We will complete an analysis using mixed effects models to predict $\tau$ and deltas from the number of chemotherapy treatments with a random intercept for patients. For aim 2, ANOVAs will be used to assess differences in mitochondrial capacity between drug regimens at each chemotherapy cycle.

## Data management plan

Patient health information (PHI) will be obtained directly from the patient and the electronic medical record (EMR). This information will only be obtained after documented informed consent has been provided by the patient. PHI will be stored in the Prisma Health RedCap˚, a HIPAA compliant cloud platform. All data will be housed within the Prisma Health system. Surveys and questionnaires will be stored in a locked file cabinet in a locked room in the Human Performance Laboratory.

## Trial oversight

This study has been approved by the Institutional Review Board at the Prisma Office of Research Compliance and Administration (ORCA). All adverse events (AE) and serious adverse events (SAE) will be recorded from the start of the visits, beginning with the first visit for lactate threshold testing and throughout completion of exercise visits, or when the participant has elected to no longer participate. The investigators will record and report all AE and SAE to the IRB within 24 hours. Fatal or life threatening unexpected adverse reactions will be reported to Prisma Health IRB immediately.

## Discussion

With limited studies of mitochondrial function (e.g., oxygen delivery and utilization) during cancer chemotherapy, opportunity exists to address mechanisms of cachexia and cancer-related fatigue using NIRS. To date, two main studies used NIRS with cancer populations. However, neither of the research teams collected data during active chemotherapy treatment, and they only measured steady-state values during exercise as opposed to collecting oxygen kinetics during recovery and exercise. Additionally, the independent variables included treatments additional to chemotherapy, including radiation and immunotherapy. The variation in independent variables may have been the reason for limited changes in the NIRS outcome variables. Ederer et al [29] found significant differences in HHb and total hemoglobin with a limited sample size of 8 cancer survivors compared to matched controls, while Grigoriadis et al. [30] found no significant differences in any of the NIRS variables. Understanding oxygen delivery and utilization kinetics with ventilatory and local NIRS measurements during chemotherapy could unlock mechanistic understanding of mitochondrial dysfunction. Our study will be the first to recruit at least 30 patients diagnosed with cancer and capture both baseline and longitudinal NIRS measurements throughout the entire chemotherapy protocol.

The study's design must consider 1) exercise tolerance levels to obtain fatigue and recovery kinetics for patients going through chemotherapy and 2) rapid recruitment of patients for baseline measurements. We recognize that the level of conditioning our patients have at baseline may be reduced as treatment progresses. Therefore, we chose a protocol that was tolerable to patients, which is to cycle to 10% above their lactate threshold level; both before discontinuing the LT test and for the on/off kinetics tests. This intensity elicits an appropriate mitochondrial response in the exercising muscle while allowing for a reasonable exercise intensity that a patient can complete despite anticipated treatment deconditioning. Additionally, because patient recruitment will occur at the time of diagnosis, a team of referral coordinators, nurse navigators, and oncologists need to identify newly diagnosed patients, and be prepared, through a "chemo-teach", to consent interested patients in a timely manner.

A few limitations are to be considered regarding the validity of the PortaMon by Artinis Medical Systems. First, the device is designed to accurately penetrate 20 mm of depth through adipose tissue and into the muscle belly. If the subcutaneous adipose tissue thickness is greater than 20 mm, the infrared signal from the device will not penetrate the skeletal muscle body and be unable to obtain accurate measurements. Our protocol minimizes this concern by excluding participants who have greater than 20 mm adipose tissue subcutaneous to the vastus lateralis, and by standardizing device placement on the leg from test to test to increase reliability and validity of repeated measures measurements. Ambient light can interfere with the signal quality, however Oxysoft provides real-time data acquisition (DAQ) measurements. Using the manufacturer's recommendation, channels with DAQ higher than 97% will be removed from analysis.

Results from this pilot study will provide information on the feasibility of capturing on/off kinetics mitochondrial function data longitudinally using NIRS in patients newly diagnosed with breast or gynecological cancer, provide preliminary data for future extramural funding, and inform the scientific community of results through dissemination via conference presentations and peer-reviewed journal publications.

### Trial status

The trial is currently in the data collection phase with 26 participants who have completed the protocol. The study is planned to continue to enroll test subjects until n = 30.

## Supporting information

**S1 Checklist. SPIRIT 2013 checklist: recommended items to address in a clinical trial protocol and related documents\*.**
(DOCX)

**S1 File. Near Infrared Spectroscopy (NIRS) as a method for measuring oxidative capacity of skeletal muscle mitochondria in breast cancer and all gynecological cancer patients protocol.**
(PDF)

## Author contributions

**Conceptualization:** Randolph Edward Hutchison, Shannon Smith, Chloe Caudell, William Larry Gluck, Jennifer L. Trilk.

**Data curation:** Randolph Edward Hutchison, Shannon Smith, Chloe Caudell, Sara Biddle.

**Formal analysis:** Randolph Edward Hutchison, Sara Biddle, Jennifer L. Trilk.

**Funding acquisition:** Randolph Edward Hutchison, Shannon Smith, Chloe Caudell, William Larry Gluck, Jennifer L. Trilk.

**Investigation:** Randolph Edward Hutchison, Chloe Caudell, William Larry Gluck, Jennifer L. Trilk.

**Methodology:** Randolph Edward Hutchison, Shannon Smith, Chloe Caudell, Sara Biddle, William Larry Gluck, Jennifer L. Trilk.

**Project administration:** Randolph Edward Hutchison, Sara Biddle, William Larry Gluck, Jennifer L. Trilk.

**Resources:** Randolph Edward Hutchison, William Larry Gluck, Jennifer L. Trilk.

**Software:** Randolph Edward Hutchison, Sara Biddle, William Larry Gluck.

**Supervision:** Randolph Edward Hutchison, Sara Biddle, William Larry Gluck, Jennifer L. Trilk.

**Validation:** Randolph Edward Hutchison, Shannon Smith, Chloe Caudell, Sara Biddle, William Larry Gluck, Jennifer L. Trilk.

**Visualization:** Randolph Edward Hutchison, Shannon Smith, Chloe Caudell, Sara Biddle, William Larry Gluck, Jennifer L. Trilk.

**Writing – original draft:** Randolph Edward Hutchison, Shannon Smith, Chloe Caudell, Sara Biddle, William Larry Gluck, Jennifer L. Trilk.

**Writing – review & editing:** Randolph Edward Hutchison, Sara Biddle, William Larry Gluck, Jennifer L. Trilk.

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
