## [Decision Letter · Decision Letter 0]

Dear Dr. Trilk,

Thank you for submitting your manuscript to PLOS ONE. After careful consideration, we feel that it has merit but does not fully meet PLOS ONE’s publication criteria as it currently stands. Therefore, we invite you to submit a revised version of the manuscript that addresses the points raised during the review process.

**ACADEMIC EDITOR: **
**Dear Authors,**

We look forward to receiving your revised manuscript.

Kind regards,

Darpan I. Patel, PhD

Academic Editor

PLOS ONE

2. We note that you have selected “Clinical Trial” as your article type. PLOS ONE requires that all clinical trials are registered in an appropriate registry (the WHO list of approved registries is at https://www.who.int/clinical-trials-registry-platform/network/primary-registries
https://www.who.int/clinical-trials-registry-platform/network/primary-registries and more information on trial registration is at http://www.icmje.org/about-icmje/faqs/clinical-trials-registration/). Please state the name of the registry and the registration number (e.g. ISRCTN or ClinicalTrials.gov) in the submission data and on the title page of your manuscript. a) Please provide the complete date range for participant recruitment and follow-up in the methods section of your manuscript. b) If you have not yet registered your trial in an appropriate registry, we now require you to do so and will need confirmation of the trial registry number before we can pass your paper to the next stage of review. Please include in the Methods section of your paper your reasons for not registering this study before enrolment of participants started. Please confirm that all related trials are registered by stating: “The authors confirm that all ongoing and related trials for this drug/intervention are registered”. Please see http://journals.plos.org/plosone/s/submission-guidelines#loc-clinical-trials for our policies on clinical trials.

Reviewers' comments:

Reviewer's Responses to Questions

**Comments to the Author**

1. Does the manuscript provide a valid rationale for the proposed study, with clearly identified and justified research questions?

Reviewer #1: Yes

Reviewer #2: Yes

2. Is the protocol technically sound and planned in a manner that will lead to a meaningful outcome and allow testing the stated hypotheses?

Reviewer #1: No

Reviewer #2: Yes

3. Is the methodology feasible and described in sufficient detail to allow the work to be replicable?

Reviewer #1: No

Reviewer #2: Yes

4. Have the authors described where all data underlying the findings will be made available when the study is complete?

Reviewer #1: Yes

Reviewer #2: Yes

5. Is the manuscript presented in an intelligible fashion and written in standard English?

Reviewer #1: Yes

Reviewer #2: Yes

You may also provide optional suggestions and comments to authors that they might find helpful in planning their study.

**Reviewer #1:**  The aims of the study are to develop a protocol that 1) assesses mitochondrial dysfunction in cancer patients using exercise that is tolerable for the patients throughout the entire chemotherapy regimen and 2) explores the link between oxygen utilization at the muscular level via NIRS with whole body traditional metabolic ventilatory measures.

It seems to be a two step process. First the investigators will use an observational mixed model repeated measures design to longitudinally compare (pre-chemotherapy and throughout treatment) patients’ skeletal muscle mitochondrial capacity of the vastus lateralis during stationary cycling. The random effect for slope and intercept are reasonable. However, some of the detail is not quite clear. What is the rationale for an unstructured correlation? Also, the eligibility criteria has to be clarified a bit more for model purposes. Will breast cancer and gynecologic subjects be analyzed separately? What specifically would be the ineligibility criteria? The independent variables are therapy and number of treatments. There are three dependent variables which will increase the number of analyses. Also , how will the covariates listed on the left side of Figures 1A and 1B be incorporated into the analysis? Will this be a repeated ANCOVA instead of an ANOVA?

What accommodation will be made in the sample size of just 30 for dropouts or missing data? One has difficulty anticipating how the analysis will be organized and reported for this endeavor? When the investigators say that, ‘We assume that at least n=8 of each chemotherapy type will be necessary to detect any differences in trajectory of τ over time’, what does that mean exactly? How many chemotherapy types are there exactly for these diagnoses?

The second step is obviously the oxygenation via NIRS which the authors explain in a separate data analysis and statistical analysis section. The organization of the presentation is challenging to navigate through. The authors perhaps should re-edit this paper more directed to presenting the material in a protocol topic format .

**Reviewer #2: ** This study has a well-designed protocol with clear goals to assess the impact of chemotherapy on mitochondrial function in cancer patients. The experimental design, equipment, and statistical analysis are appropriate for the research objectives. However, several points need to be clarified before being considered for publication:

1) I question the necessity of explicitly stating that the results will be shared through conferences and publications in the last sentence of the abstract. But this made me wonder if your results are likely not going to be published ? If so this should be addressed in the manuscript for transparency. Since “Data collection started on July 26, 2021 and is ongoing through May 1, 2025.” but only 13 out of the 30 planned participants have been enrolled so far, it seems unlikely the enrollment goal will be reached.

2) Weight loss or cachexia status is not explicitly mentioned as a tracked variable. Not all cancer patients and not all patients undergoing chemotherapy develop cachexia. Some may maintain weight, while others experience severe muscle wasting and cachexia itself affects mitochondrial function, independent of chemotherapy, potentially confounding the study’s main outcome. The manuscript needs to mention this and separate analyses for cachectic vs. non-cachectic patients should be ran to determine whether cachexia modifies the effect of chemotherapy on mitochondrial function.

3) The introduction contains becomes difficult to follow due to several flow andgrammatical issues, or incomplete sentences and erroneous statements. The entire manuscript should be carefully reviewed for proper language. For example:

- cachexia and chemotherapy-induced muscle wasting are related but distinct conditions and should not be distinguished (line79: “chemotherapy-induced muscle wasting, also known as cachexia”).

- Line 84-86 : "therefore, the increase in survivorship has been an increase in focus on understanding mechanisms of cachexia, such as mitochondrial dysfunction, with the aim of studying these mechanisms to understand how to improve quality of life for [10]."

- Line 88-90 “Muscle biopsies have been the gold standard assessment but cause significant discomfort to the patient and ma y not represent in vivo activation of the skeletal working muscle tissue associated with [17].”

- Line 109-110: “assessment of age-related differences in skeletal muscle oxidative [17].”

- Line 236-239 : “Pictures of the anatomical location of NIRS device placement with a tape measure from the center of the anterior border of the patella to the top of the NIRS device will be taken to standardize placement at every visit.”

**Do you want your identity to be public for this peer review?** For information about this choice, including consent withdrawal, please see our Privacy Policy

Reviewer #1: No

Reviewer #2: No

---

## [Author Response · Author response to Decision Letter 1]

17 May 2025

Thank you for the opportunity to improve our manuscript. We have edited the manuscript to be in line with the PLOS ONE’s style requirements.

2. We note that you have selected “Clinical Trial” as your article type. PLOS ONE requires that all clinical trials are registered in an appropriate registry (the WHO list of approved registries is at https://www.who.int/clinical-trials-registry-platform/network/primary-registries
https://www.who.int/clinical-trials-registry-platform/network/primary-registries and more information on trial registration is at http://www.icmje.org/about-icmje/faqs/clinical-trials-registration/). Please state the name of the registry and the registration number (e.g. ISRCTN or ClinicalTrials.gov) in the submission data and on the title page of your manuscript. a) Please provide the complete date range for participant recruitment and follow-up in the methods section of your manuscript. b) If you have not yet registered your trial in an appropriate registry, we now require you to do so and will need confirmation of the trial registry number before we can pass your paper to the next stage of review. Please include in the Methods section of your paper your reasons for not registering this study before enrolment of participants started. Please confirm that all related trials are registered by stating: “The authors confirm that all ongoing and related trials for this drug/intervention are registered”. Please see http://journals.plos.org/plosone/s/submission-guidelines#loc-clinical-trials for our policies on clinical trials.

Thank you for the opportunity to improve our manuscript. The following sentences are in lines 147-149 in the original manuscript and have been added to the title page in the revised manuscript (lines 28-29). “This clinical trial has been registered with clinicaltrials.gov (Identifier: NCT006672497). Data collection started on July 26, 2021 and is ongoing through May 1, 2025.” The additional sentence, “The authors confirm that all ongoing and related trials for this observation trial (no drug or intervention) are registered.” has been added to the title page and in lines 152-153 of the revised manuscript.

Reviewer #1: The aims of the study are to develop a protocol that 1) assesses mitochondrial dysfunction in cancer patients using exercise that is tolerable for the patients throughout the entire chemotherapy regimen and 2) explores the link between oxygen utilization at the muscular level via NIRS with whole body traditional metabolic ventilatory measures.

It seems to be a two-step process. First the investigators will use an observational mixed model repeated measures design to longitudinally compare (pre-chemotherapy and throughout treatment) patients’ skeletal muscle mitochondrial capacity of the vastus lateralis during stationary cycling. The random effect for slope and intercept are reasonable. However, some of the detail is not quite clear. What is the rationale for an unstructured correlation?

Many thanks to the reviewer for comments to improve our manuscript. Based on the work of Cheng et al. [35], they suggest that with a “fixed effect of time (chemotherapy cycle), “time may be included as a categorical predictor in the fixed and random effects, which corresponds exactly to a fully saturated polynomial design matrix in time. In such cases, a wide range of covariance structures has appeal, especially unstructured patterns. Similar to the selection of fixed effects, the evaluation of covariance structure should depend on scientific knowledge in the specific area as well as statistical criteria.” Since this is the first study to measure mitochondrial function during chemotherapy, there is little to no evidence to support another approach. See other references (33-36 for further support. We have added two sentences to elaborate on this rationale per the reviewer’s comments in line (171-172).

Also, the eligibility criteria has to be clarified a bit more for model purposes. Will breast cancer and gynecologic subjects be analyzed separately? What specifically would be the ineligibility criteria?

Thank you for the comments to improve our manuscript. Because the treatment for the gynecological patients and the breast cancer patients is very similar per NCCN guidelines, both populations will be analyzed together. In reality, they are all women and all adjuvant. At the time of administration, there is no clinical evidence of disease (non-metastatic). Therefore, the patients that come in are similar from a population viewpoint. The regimens have much the same overlap. It’s disease “agnostic”. The drugs/treatment regimens are much the same. If the treatment is significantly different based on their oncologist’s recommendation, then the populations would be analyzed separately. We have added 2 sentences (lines 192-193) to clarify the manuscript.

The independent variables are therapy and number of treatments. There are three dependent variables which will increase the number of analyses.

Also, how will the covariates listed on the left side of Figures 1A and 1B be incorporated into the analysis? Will this be a repeated ANCOVA instead of an ANOVA?

Thank you for the comments to improve our manuscript. In figure 1A and 1B, the additional outcome measures are being collected as additional whole-body vs. muscle level cardio-metabolic health. With 30 subjects, we do not anticipate having enough data points to sub-divide the data by these measures to have meaningful analysis. Meaning, the statistical assumptions would not be able to be met. This additional data could be considered more holistically in a clinical case study. If warranted, another future study could collect more data to statistically analyze with these as true covariates.

What accommodation will be made in the sample size of just 30 for dropouts or missing data?

Thank you for the comments to improve our manuscript. We have added a sentence (lines 180-182) stating that data collection will continue until we reach 30 subjects who have completed the study, excluding dropouts or subjects whose data was unusable due to missing data.

One has difficulty anticipating how the analysis will be organized and reported for this endeavor? When the investigators say that, ‘We assume that at least n=8 of each chemotherapy type will be necessary to detect any differences in trajectory of τ over time’, what does that mean exactly? How many chemotherapy types are there exactly for these diagnoses?

Thank you for the comments and opportunity to improve the manuscript. We have revised lines 182-191 and added additional sentences to reflect what we have seen in pilot data so far and in conversations with several of the oncologist in the Cancer Institute. Based on the NCCN guidelines, treatment can be considered cytotoxic (platinum based, anthracyclines, etc.) or non-cytotoxic such as immunotherapy. These treatments can be given separately or at the same time. Generally, they consist of 3 or more cycles. Therefore, we have added revisions to reflect that differences between cytotoxic and non-cytotoxic treatment could be considered based on current NCCN guidelines. If the NCCN guidelines change based on research or development of new drug regimens, comparisons, could also change.

The second step is obviously the oxygenation via NIRS which the authors explain in a separate data analysis and statistical analysis section. The organization of the presentation is challenging to navigate through. The authors perhaps should re-edit this paper more directed to presenting the material in a protocol topic format .

Thank you for the comments to improve our manuscript. We have revisited the PLOS One Study Protocol Article Template to make sure we are following the proper format. We have edited the methods as well as added several sentences throughout for clarification and to improve readability.

Reviewer #2: This study has a well-designed protocol with clear goals to assess the impact of chemotherapy on mitochondrial function in cancer patients. The experimental design, equipment, and statistical analysis are appropriate for the research objectives. However, several points need to be clarified before being considered for publication:

1) I question the necessity of explicitly stating that the results will be shared through conferences and publications in the last sentence of the abstract. But this made me wonder if your results are likely not going to be published ? If so this should be addressed in the manuscript for transparency.

Thank you for the comments to improve our manuscript. We have every intention of publishing our results in peer-reviewed journals in the field. We have added the word “journal” to the last sentence of the abstract (line 59 revised manuscript) and line 406 in the last sentence of the discussion.

Since “Data collection started on July 26, 2021 and is ongoing through May 1, 2025.” but only 13 out of the 30 planned participants have been enrolled so far, it seems unlikely the enrollment goal will be reached.

Thank you for the comments to improve our manuscript. We are on track for meeting our goal of 30 patients and have completed testing of 26 out of 30 patients at the time of these revisions. This change has been reflected in line 583 of the revised manuscript. It is very common with patient recruitment to have a slower rate than other normal human subject studies. Patient recruitment with clinical trials, especially with cancer patients can sometimes be as low as 2-4 patients per year. We have extended the end date of the IRB for recruitment to March 27, 2026 and reflected this in revising line 152.

2) Weight loss or cachexia status is not explicitly mentioned as a tracked variable. Not all cancer patients and not all patients undergoing chemotherapy develop cachexia. Some may maintain weight, while others experience severe muscle wasting and cachexia itself affects mitochondrial function, independent of chemotherapy, potentially confounding the study’s main outcome. The manuscript needs to mention this and separate analyses for cachectic vs. non-cachectic patients should be ran to determine whether cachexia modifies the effect of chemotherapy on mitochondrial function.

Thank you for this observation and comments to improve our manuscript. As a pilot observational study, our primary outcome measure is reduction in mitochondrial function of patients going through chemotherapy. The mechanism of why there is mitochondrial dysfunction is multi-factorial including effects from cachexia, but it is not our intention to elucidate this underlying mechanism, only the outcome of reduced mitochondrial capacity during chemotherapy. Chemotherapy induced mitochondrial dysfunction has a more rapid effect vs. the slower effect of cachexia. Our primary measurements do not include the measurement of cachexia. However, we do understand that the wording in the manuscript may confuse readers, therefore have improved wording that streamlines our “if/then” algorithm. While chemotherapy can induce cachexia, it does have a component of severe inflammatory skeletal muscle response. Much of this is due to mitochondrial dysfunction. That being said, we agree with the reviewer that the patient may or not have cachexia and have clarified wording in the manuscript concerning cachexia in lines 85-93.

3) The introduction contains becomes difficult to follow due to several flow and grammatical issues, or incomplete sentences and erroneous statements. The entire manuscript should be carefully reviewed for proper language.

Thank you for this comment. Introduction has been edited to reflect this change.

For example:

- cachexia and chemotherapy-induced muscle wasting are related but distinct conditions and should not be distinguished (line79: “chemotherapy-induced muscle wasting, also known as cachexia”).

Thank you for this comment. Line 79 has been edited to reflect this change.

- Line 84-86 : "therefore, the increase in survivorship has been an increase in focus on understanding mechanisms of cachexia, such as mitochondrial dysfunction, with the aim of studying these mechanisms to understand how to improve quality of life for [10]."

Thank you for this comment. Lines 84-86 have been edited to reflect this change.

- Line 88-90 “Muscle biopsies have been the gold standard assessment but cause significant discomfort to the patient and ma y not represent in vivo activation of the skeletal working muscle tissue associated with [17].”

Thank you for this comment. Lines 88-90 have been edited to reflect this change.

- Line 109-110: “assessment of age-related differences in skeletal muscle oxidative [17].”

Thank you for this comment. Lines 109-110 have been edited to reflect this change.

- Line 236-239 : “Pictures of the anatomical location of NIRS device placement with a tape measure from the center of the anterior border of the patella to the top of the NIRS device will be taken to standardize placement at every visit.”

Thank you for this comment. Lines 236-239 (lines 250, 259-260 in revised manuscript) have been edited to reflect this change.

7. PLOS authors have the option to publish the peer review history of their article (what does this mean?). If published, this will include your full peer review and any attached files.

Do you want your identity to be public for this peer review? For information about this choice, including consent withdrawal, please see our Privacy Policy.

Reviewer #1: No

Reviewer #2: No

In compliance with data protection regulations, you may request that we remove your personal registration details at any time. (Remove my information/details). Please contact the publication office if you have any questions.Error! Filename not specified.

Response:

The authors thank the reviewers for their constructive comments.

We have addressed or commented on all of the points presented, and have compiled a summary of the changes and responses below for reconsideration. The original review text is in pink, and our replies are in black.

---

## [Decision Letter · Decision Letter 1]

Effects of chemotherapy on skeletal muscle mitochondrial oxidative capacity using near-infrared spectroscopy (NIRS): Protocol paper for an observational mixed model repeated measures design in patients with breast and gynecological cancer

PONE-D-24-49905R1

Dear Dr. Trilk,

We’re pleased to inform you that your manuscript has been judged scientifically suitable for publication and will be formally accepted for publication once it meets all outstanding technical requirements.

Kind regards,

Darpan I. Patel, PhD

Academic Editor

PLOS ONE

Reviewer's Responses to Questions

**Comments to the Author**

1. Does the manuscript provide a valid rationale for the proposed study, with clearly identified and justified research questions?

Reviewer #1: Yes

Reviewer #2: Yes

2. Is the protocol technically sound and planned in a manner that will lead to a meaningful outcome and allow testing the stated hypotheses?

Reviewer #1: Yes

Reviewer #2: Yes

3. Is the methodology feasible and described in sufficient detail to allow the work to be replicable?

Reviewer #1: Yes

Reviewer #2: Yes

4. Have the authors described where all data underlying the findings will be made available when the study is complete?

Reviewer #1: Yes

Reviewer #2: Yes

5. Is the manuscript presented in an intelligible fashion and written in standard English?

Reviewer #1: Yes

Reviewer #2: Yes

You may also provide optional suggestions and comments to authors that they might find helpful in planning their study.

Reviewer #1: All comments have been addressed and the revisions included in the manuscript.

Reviewer #2: The study by Hutchison et al. proposes a protocol to assess the effects of chemotherapy on mitochondrial function in cancer patients, using a NIRS and a patient-tolerable exercise protocol. In the revised manuscript, the authors addressed the previous concerns regarding clarity, structure, and methodology. The manuscript now presents a clear and viable study.

**Do you want your identity to be public for this peer review?** For information about this choice, including consent withdrawal, please see our Privacy Policy

Reviewer #1: No

Reviewer #2: No

---

## [Editor Report · Acceptance letter]

PONE-D-24-49905R1

PLOS ONE

Dear Dr. Trilk,

I'm pleased to inform you that your manuscript has been deemed suitable for publication in PLOS ONE. Congratulations! Your manuscript is now being handed over to our production team.

Kind regards,

on behalf of

Dr. Darpan I. Patel

Academic Editor

PLOS ONE